# Synthesis and Physicochemical Properties of Poly(vinyl) Alcohol Nanocomposites Reinforced with Nanocrystalline Cellulose from Tea (*Camellia sinensis*) Waste

**DOI:** 10.3390/ma14237154

**Published:** 2021-11-24

**Authors:** Fauzi Handoko, Yusril Yusuf

**Affiliations:** Department of Physics, Faculty of Mathematics and Natural Sciences, Universitas Gadjah Mada, Yogyakarta 55281, Indonesia; fauzihandoko@mail.ugm.ac.id

**Keywords:** tea waste, nanocrystalline cellulose, poly(vinyl) alcohol, nanocomposites

## Abstract

The purpose of this study was to utilize cellulose from tea waste as nanocrystalline cellulose (NCC), which is used as a filler in poly(vinyl) alcohol (PVA) nanocomposites. To obtain the NCC, a chemical process was conducted in the form of alkali treatment, followed by bleaching and hydrolysis. Nanocomposites were formed by mixing PVA with various NCC suspensions. With chemical treatment, lignin and hemicellulose can be removed from the tea waste to obtain NCC. This can be seen in the functional groups of cellulose and the increase in crystallinity. The NCC had a mean diameter of 6.99 ± 0.50 nm. Furthermore, the addition of NCC to the PVA nanocomposite influenced the properties of the nanocomposites. This can be seen in the general increase in opacity value, thermal and mechanical properties, and crystallinity, as well as the decrease in the value of the swelling ratio after adding NCC. This study has revealed that NCC from tea waste can be used to improve the physicochemical properties of PVA film.

## 1. Introduction

Polymer nanocomposite films comprise nanofillers dispersed in a polymer matrix. Incorporating some nano-sized fillers can improve the composite properties required for many industrial and technological applications. Polymer nanocomposite films with inorganic fillers can improve stiffness, strength, hardness, and high temperature creep resistance compared to unfilled polymers [1,2,3]. These nanocomposite films have recently become an issue of great concern from an economic, environmental, and performance point of view. This can be overcome by replacing inorganic fillers with natural materials [4,5,6].

The organic materials that are often used are natural fibers. Natural fibers generally come from plants and animals. Plant fiber can be further classified into subgroups according to its source, for example, stem, leaf, seed, or fruit. The main component of plant fiber is cellulose, which is a naturally occurring hydrophilic polymer.

Cellulose is a natural biopolymer that is renewable and biodegradable. Cellulose comprises two glucose molecules linked by a -1,4-glycosidic bond [7,8]. Insulating cellulose fibers and reinforcing polymers with them to make nanocomposite films is very advantageous. Fibrils are defined as small, slender fibers or filaments. These may include cellulose crystals or whiskers, and microfibrils [9]. 

Using various methods, e.g., physical treatments, chemical treatments, and biological treatments, it is possible to obtain cellulose with nanometric dimensions called nanocellulose [10]. Nanocellulose has interesting characteristics; it is lightweight, has high mechanical properties, a high length-to-diameter ratio, and a large specific surface area [11]. Based on ISO (International Organization for Standardization) standards (ISO TS 20477: 2017), nanocellulose can be divided into two categories. The first is cellulose nano-objects, such as cellulose nanocrystals (CNC) and cellulose nanofibrils (CNF). The second type of nanocellulose is nanostructured cellulose, which includes cellulose microcrystals (CMC), microcrystalline cellulose (MCC), cellulose microfibrils (CMF), microfibrillated cellulose (MFC), and bacterial cellulose (BC) [12].

Chemical methods, such as strong acid hydrolysis, remove amorphous regions from cellulose fibers and produce nano-sized fibers. Cellulose contains both crystalline and amorphous regions. The crystalline particles obtained after removing the amorphous region via acid hydrolysis treatment are termed nanocrystalline cellulose (NCC). The acid hydrolysis process is conducted using a strong acid solution, such as sulfuric [13] or hydrochloric acid [14]. Because of its superior surface activity and large specific surface area, the addition of NCC can improve the mechanical properties of composites [15]. Oil palm empty fruit bunch pulp [9], bamboo fiber [16], olive fiber [17], pineapple crown waste [18], and tea [19] were used as raw materials to isolate the NCC.

Indonesia is among the world’s biggest tea producers. According to the Statista Research Department, in 2020, approximately 127.9 thousand metric tons of tea were produced in Indonesia. A large amount of tea waste is generated during tea processing. Currently, most tea waste is only used for composting or in landfills. However, it has put pressure on the environment because tea waste is difficult to decompose and causes a huge loss of useful components [19]. Therefore, the use of tea waste is currently receiving considerable attention. Tea waste reportedly contains 16.2% cellulose, 68.2% hemicellulose, and 18.8% lignin [20]. Thus, tea waste can be used as a raw material in the manufacture of NCC, which will be a nanofiller in polymer nanocomposite films.

Among various polymers, poly(vinyl) alcohol (PVA) is studied here because it has the specific properties, such as water solubility, semi-crystallinity, non-toxicity, transparency, biocompatibility, and biodegradability [4]. PVA has good chemical and physical stability, and can form fibers, films, and membranes [21]. Due to its excellent film-forming properties in aqueous solvents, PVA-based films can be easily fabricated, and the prepared films have excellent tensile strength, flexibility, and oxygen barrier properties. Thus, PVA-based films have been interesting to observe. However, PVA has poor resistance to wet environments due to its hydrophilic nature, which is caused by the presence of abundant hydroxyl groups. In an aqueous environment, water molecules can easily penetrate the PVA film, causing swelling and thereby destroying its mechanical properties [22]. To overcome the disadvantages of PVA films, nanofiller-based fillers are added. Hence, the combined advantages of nanofillers should be explored.

In this study, the tea waste used to produce NCC is an original and natural material from Indonesia. NCC can be used as filler in nanocomposite films to be applied in the food packaging, pharmaceutical, automobile, and construction industries. That PVA could be manufactured from non-petroleum-based resources has increased worldwide concerns for developing eco-friendly composites. The development of NCC for the reinforcement of nanocomposite films has gained the attention of researchers. The application of nanocomposite films could aid in mitigating environmental impact via diminished dependency on non-renewable products which are derived from fossil fuel resources, such as petroleum and natural gas. With developed PVA–NCC nanocomposite films, the amounts of undegradable residues released into nature would ultimately be reduced.

In this work, NCC was isolated using acid hydrolysis from tea waste. The NCC obtained was used as a nanofiller in manufacturing nanocomposite films. The nanocomposite films were made using a PVA matrix. PVA–NCC nanocomposite films were prepared using the casting method. Thus, in this study, the effects of adding NCC to PVA nanocomposite films were observed on the physicochemical properties. Additionally, the functional groups, morphologies, transparencies, structure crystallinity, mechanical properties, thermal properties, and swelling ratios of PVA–NCC nanocomposite films were also observed.

## 2. Materials and Methods

### 2.1. Materials

Tea waste was obtained from the production of Perseroan Terbatas Pagilaran, Batang, Central Java, Indonesia. Poly(vinyl) alcohol (PVA)—with a molecular weight of 145,000, sodium hydroxide (NaOH), hydrogen peroxide 30% (H_2_O_2_), glacial acetic acid (CH_3_COOH), and hydrochloric acid 37% (HCl)—was purchased from Merck, Germany. In addition, sodium hypochlorite 12% (NaClO) was obtained from Brataco.

### 2.2. The Extraction of NCC

NCC was prepared using acid hydrolysis. Tea waste was sifted using a 100 mesh. The tea waste was treated using a solution of 6% (*w*/*v*) NaOH at 80 °C for 3 h. This process was repeated three times, and the solution was washed with distilled water until it obtained a neutral pH. The first bleaching process was conducted using a solution of 2.5% (*v*/*v*) NaClO, coupled with an acetate buffer (2.7 g NaOH and 7.5 mL of glacial acetic acid in 100 mL distilled water) [20] stirred at 70 °C for 1 h, and then washed with distilled water until it obtained a neutral pH. The second bleaching process was conducted using a 7% (*v*/*v*) solution of H_2_O_2_ mixed with a solution of 4% (*w*/*v*) NaOH, stirred at 70 °C for 2 h [4], and then washed with distilled water until it obtained a neutral pH. Acid hydrolysis was conducted using a solution of 5M HCl at 50 °C for 12 h [23], and then washed with distilled water until it obtained a neutral pH and NCC was obtained from tea waste in suspension.

### 2.3. Preparation of PVA–NCC Nanocomposite Films 

A 10% (*w*/*v*) PVA solution was made by dissolving 1.111 g of PVA in a 10 mL mixture of distilled water and NCC suspension and stirring at 1500 rpm at 80 °C for 3 h. Subsequently, the mixture was formed and dried in an oven at 50 °C for 24 h. The solvent used was a mixture of distilled water and NCC suspensions (Table 1).

### 2.4. Characterizations 

#### 2.4.1. Fourier Transform Infrared Spectroscopy (FTIR) 

FTIR (Thermo Nicolet iS10, Tokyo Japan) analysis was used to observe changes in the functional group at each treatment, and changes in the composition of NCC when added to nanocomposite films. The wavenumbers used were between 400 cm^−1^ and 4000 cm^−1^.

#### 2.4.2. Raman Spectroscopy

The Raman spectra of the nanocomposite films were measured using a Raman spectrometer (Thorlabs CCS100, Newton, NJ, USA) with an excitation laser of 532 nm and a resolution of 10 cm^−1^. The wavenumbers used were between 400 cm^−1^ and 4000 cm^−1^.
(1)$$\mathrm{Degree}\;\mathrm{of}\;\mathrm{crystallinity}\;\mathrm{of}\;\mathrm{PVA}\;=\frac{I_{\left(O-H\right)}}{I_{\left(C-H\right)}}$$
where $I_{\left(O-H\right)}$ is the intensity of O-H, and $I_{\left(C-H\right)}$ is the intensity of C-H.

#### 2.4.3. Morphology Analysis

The influence of each NCC manufacturing process and the effect of adding NCC to nanocomposite films was analyzed using a scanning electron microscope (Jeol JSM-6510LA, Tokyo, Japan) with an accelerating voltage of 10–15 kV. The samples were mounted on metal studs using double-sided adhesive tapes. The samples were coated using platinum to avoid charging during the tests. The diameter size and morphology of the NCC were determined using a transmission electron microscope (Jeol Jem-1400, Tokyo, Japan). The diameter size distribution of the NCC was calculated using ImageJ software version 2006 (National Institutes of Health (NIH), Bethesda, MD, USA).

#### 2.4.4. Crystallographic Analysis 

The crystallinity of the nanocomposite films was measured using an X-ray diffractometer (PANalytical Type X’Pert Pro, Tokyo, Japan) with Cu-Kα radiation at λ = 0.154 nm. The XRD data were taken in the range of 2θ = 5–60°. The crystalline index (CI) was calculated using Equation (2) [24]:(2)$$CI=\frac{I_{002}-I_{am}}{I_{002}}\times 100\;$$
where *I_002_* is the maximum intensity of the crystalline regions, and *I_am_* is the lowest intensity of the amorphous regions in the sample. The degrees of crystallinity of the nanocomposite films were calculated using Herman’s equation, and the CI was calculated using Equation (3) [25]:(3)$$CI=\frac{A_{cr}}{A_{t}}\times 100$$
where *A_cr_* is the crystalline area, and *A_t_* is the total area of the diffractogram.

#### 2.4.5. Analysis of Thermal Properties

The thermal properties of the nanocomposite films were assessed using differential scanning calorimetry (Shimadzu DSC 60 Series, Tokyo, Japan). A total of 10 mg of the sample was prepared, placed in an aluminum container, and pressed with the aluminum container into pellets. The device was set at 30 °C–400 °C with a 10 °C/min rise in nitrogen atmosphere. 

#### 2.4.6. Transparency of the Nanocomposite Film

The transparency of the nanocomposite films was measured using a UV-Vis spectrophotometer (Ocean Optics USB4000, Dunedin, FL, USA) via the ASTM standard D1003-00 method, with a transmittance between 400 nm and 800 nm. Measurement was repeated three times [23].

#### 2.4.7. Swelling Ratios of the Nanocomposite Film

The swelling behaviors of the nanocomposite films were observed by cutting the nanocomposite into a square shape, sized 10 mm × 10 mm. Subsequently, the initial weight (*W_0_*) was measured before the sample was soaked in distilled water for 24 h. The sample was drained slowly on filter paper to remove excess water from the nanocomposite’s surface, and the sample was weighed when wet (*W*). The swelling ratio was calculated using Equation (4) [26]:(4)$$Ratio\;swelling=\left(\frac{W-W_{0}}{W_{0}}\right)\times 100$$

The experiment was repeated three times.

#### 2.4.8. Analysis of Mechanical Properties

The mechanical properties (tensile strength, Young’s modulus, and elongation at break) of the nanocomposite films were observed using a universal testing machine (Zwick DO-FB.5TS, Kennesaw, GA, USA). Samples were cut following ASTM standard D638 type V.

#### 2.4.9. Statistical Analysis

The statistical significance of the tensile strength, elastic modulus, elongation at break, opacity, and swelling ratio values for nanocomposite films were verified using mean ± standard deviation (SD) and one-way variant analysis (ANOVA). The Tukey method was used to test mean differences. Statistical significance was considered as *p* < 0.05. 

## 3. Results

### 3.1. FTIR and Raman Analysis

The FTIR spectra of any chemical treatment on tea waste and nanocomposite films are shown in Figure 1a,b. The FTIR spectra of any chemical treatment of tea waste (Figure 1a) showed a broad peak at 3408 cm^−1^ due to the bonded O-H group in the cellulose molecules [27]. The peak intensity of the O-H group was found to be lower in the tea waste due to the presence of non-cellulose components that were still contained. Then, the peak intensity of the O-H group increased after alkali treatment, bleaching, and hydrolysis, which indicated an increased cellulose content. The peak at 2922 cm^−1^ was related to the stretching vibration of the C-H produced by the methyl, and methyl groups, in cellulose, hemicellulose, and lignin [28]. The peak at 898 cm^−1^ corresponds to the –CH_2_ of glycosidic β in cellulose [20]. The alkali treatment was performed to remove remnants of lignin and hemicellulose, as evidenced by the decreasing transmittance intensity at 898 cm^−1^. A stretching vibration related to C-O was observed at 1033 cm^−1^ [29]. The peak at 1047 cm^−1^ was related to the stretching vibration of the C-O-C from both the hemicellulose and cellulose [18,30]. The C=C stretching vibration at 1638 cm^−1^ indicates an aromatic group in lignin still present in tea waste [19]. The transmittance peak intensity at 1638 cm^−1^ was decreased, indicating that the lignin content was minimized after chemical treatment. 

Figure 1b shows the FTIR spectra of the nanocomposite films. The peak at 3419 cm^−1^ was related to the stretching vibration of the O-H from the intermolecular and intramolecular hydrogen bonds between the hydroxyl groups of PVA and NCC (Figure 2) [31]. Adding NCC to the PVA matrix generates a much sharper stretching peak with higher intensity, presumably due to the contribution of O-H groups from NCC. In fact, interaction through hydrogen bonding involving NCC and the PVA matrix is likely to contribute to the broadening of the O-H band. The presence of NCC in the nanocomposite films shifts this band, presumably due to intermolecular hydrogen bonding interactions with O-H groups on the surface of NCC [32]. The peak at 2922 cm^−1^ was related to the stretching vibrations of the C-H, which is an aliphatic cluster in nanocomposite films [33]. It decreased in intensity as the NCC increased, which can be explained by the contribution of the C–H vibration from NCC in the nanocomposite films. The peak at 1737 cm^−1^ was related to the C=O of acetate group residues in the PVA matrix [34]. The C=O stretching vibration was present in the NCC, thus confirming the reaction between the NCC and the PVA. The peak at 1085 cm^−1^ represents the C-O-C stretching of the pyranose ring derived from NCC. This peak was formed when NCC was added to the PVA matrix [11]. However, the intensity of that peak was reduced when NCC was added.

The Raman spectroscopy of the nanocomposite films is shown in Figure 1c. The peaks at 2929 cm^−1^, 1453 cm^−1^, 930 cm^−1^, and 854 cm^−1^ are characteristic of PVA. There was a decrease in peak size at 2929 cm^−1^ after adding NCC. This indicated that the chain structure of PVA changed with the addition of NCC. These changes were the result of the intermolecular complex formed due to the hydrogen bonds between PVA and NCC [25]. The peak at 1094 cm^−1^ indicated the C-O stretching and O-H bending from the amorphous sequence of PVA. The peak at 854 cm^−1^ indicated -CH_2_ rocking in the PVA. The peak at 1094 cm^−1^ was associated with the PVA amorphous regions, and this peak was influenced by the addition of NCC. Variations in the intensity ratio of $I_{\left(O-H\right)}/I_{\left(C-H\right)}$ indicated the degree of PVA crystallinity. There was an increase and then a decrease in the intensity ratio upon adding NCC (Figure 1d). The degree of crystallinity of the PVA–NCC nanocomposite was reduced by the rate of NCC reinforcement. This was due to the limited mobility of the PVA molecular chain in nanocomposite films [32].

### 3.2. Morphology Analysis 

The morphology of any chemical treatment of tea waste and PVA–NCC nanocomposite films is shown in Figure 3, Figure 4 and Figure 5. Figure 3a shows that the morphology of the tea waste is not uniform; some particles are in the form of long fibers and irregular shapes. This can happen because tea is produced from the leaves and stems of the tea plant. Cellulose in tea waste is still encased by non-cellulose components, such as lignin and hemicellulose. In the alkali treatment process, non-cellulose components begin to decompose. Lignin and hemicellulose decompose due to alkali treatment (black circles in Figure 3b). The sample surface became more irregular and flaky after alkali treatment [13]. This process helps with fiber decomposition and defibrillation [35]. The bleach-treated tea waste demonstrated smooth, clear, and individualized rod-like fibers of cellulose (white arrow in Figure 3c), which is indicative of the complete removal of non-cellulose components [36].

The NCC can be decomposed through acid hydrolysis (black circle in Figure 4a). This can occur when strong acids are used during acid hydrolysis [37]. Acid hydrolysis allows for the elimination of the amorphous area of tea waste cellulose fibers by unraveling the cellulose microfibril of tea waste into NCC (white arrow in Figure 4a). Utilizing acid hydrolysis significantly alters the cellulose content of tea waste. The shape of the NCC particles is thought to be due to an imperfect cooling process (quenching) during the cessation of acid hydrolysis [20]. The rest of the acid hydrolysis solution still reacts with tea waste cellulose during the cooling process, thus changing the shape from that of NCC fibers to a nanoparticle shape. During this process, there is also an agglomeration (black arrow in Figure 4a). This agglomeration possibly occurs due to the charge of ions on the surface of the crystalline area as a result of acid hydrolysis. Figure 4b shows the distribution of the sizes of the NCC; diameters ranged from 3.63 nm to 14.95 nm, with an average of 6.99 ± 0.50 nm.

The morphology of the nanocomposite films is shown in Figure 5. The PVA film surface looks smooth and uniform before the NCC suspension is added, and there is an increase in surface roughness after the addition of NCC [38]. The even and uniform distribution of NCC in the matrix can improve the mechanical properties of nanocomposite films (white arrow in Figure 5). However, when the NCC concentration was increased, small particles appeared in the PVA composite film due to agglomeration. This can be seen clearly in the PVA nanocomposite film. The NCC distribution in the PVA matrix became increasingly uneven, resulting in a large decrease in light transmission and a negative effect on the mechanical properties of the PVA–NCC nanocomposite film. This was demonstrated in the UV-Vis spectroscopy and tensile test results [39].

### 3.3. Crystallographic Analysis 

The degree of crystallization after each chemical treatment of tea waste, and the influence of NCC on the crystal structure of the nanocomposite films, were determined. Figure 6a shows the effect of the chemical treatment of tea pulp on the XRD peak. Peaks of 15.2°, 22°, and 34.4° with fields (110), (200), and (004) indicate cellulose β phases [31,40]. The crystalline structure of the NCC was at about 2θ = 22°, and the amorphous area was at about 2θ = 15.2°. Table 2 shows the crystallinity of tea waste at different stages of chemical treatments. Increased crystallinity indicates that the lignin and hemicellulose content in each treatment was reduced [41]. Materials with high crystallinity tend to have high mechanical properties, and this is important for applications in amplifiers based on nanocomposites [18]. 

Figure 6b shows the crystalline structure of the nanocomposite films. The peak characteristic of PVA at 19.4° demonstrates that the field (110) of the partially hydrolyzed PVA semi-crystalline region has a lower crystallinity compared to pure PVA crystals. The addition of NCC into the PVA nanocomposite film slightly reduced the intensity of the field (110). The crystallinity of the PVA film increases as the NCC is added [42]. Crystalline nanocomposite films are shown in Table 3. The maximum crystallinity was 39.5%, as seen in PNC1. The increased crystallization due to the bonding of NCC and PVA was evenly distributed [43]. A decreased crystallinity indicates that some hydrogen bonds between PVA chains are substituted with PVA–NCC hydrogen bonds. NCC has a higher crystallinity compared to PVA; therefore, by increasing the quantity of NCC, the crystallinity of the nanocomposite films also increases [44].

### 3.4. Analysis of Thermal Properties

The thermal properties of the nanocomposite films are presented in Figure 7. The melting point, degradation, glass transition temperatures, and enthalpy of the nanocomposite films are shown in Table 3. The peak at 100–150 °C shows an endothermic peak, which indicated that the heat energy was used to evaporate the water contained in the nanocomposite films. The nanocomposite films have two melting points, namely T_m1_ and T_m2_. T_m1_ is the melting point of PVA film. T_m1_ was between 223.17 °C and 227.32 °C. The effect of adding the NCC, which lowers the value of T_m1_, can be inferred from the results of the DSC. In PNC0, the melting point of the PVA was 227.32 °C, and after the addition of NCC, T_m1_ decreased. This can occur due to the strong interaction between the NCC and the PVA matrix, which damages the molecular chain structure. The reduction in the melting point can also result from the strong interaction between the polymer matrix and the hydroxyl group of the NCC [11]. T_m2_ is the nanocomposite film degradation. Adding NCC generally increases T_m2_. At T_m2_, the temperature exceeds 300 °C, indicating that the NCC in nanocomposite films begins to degrade [4]. 

The addition of NCC also increased the melt enthalpies (ΔH_m_) of nanocomposite films, indicating that the addition of NCC improves the crystal structure of PVA [45]. This is because the addition of NCC damages the crystalline structure of the PVA. The chain interactions in PVA and NCC were weakened, and the crystallinity of the nanocomposite films decreased.

### 3.5. Transparency of the Nanocomposite Films

The change in the transparency of the nanocomposite films upon adding NCC is shown in Figure 8a. The increasing content of NCC in PVA film decreased transparency and increased opacity (Table 3) [46]. The opacity values obtained ranged from PNC0 (12.62 au.nm) to PNC6 (307.79 au.nm). This was confirmed using a UV-Vis spectroscopy (Figure 8b), with a lower transmittance value in the nanocomposite films than in the PVA film, but with the same wavelength [14]. The transmittance value (Figure 8c) of the nanocomposite films was influenced by NCC distribution factors, such as the occurrence of agglomeration. This is consistent with the SEM test results. The addition of NCC resulted in the scattering of light, resulting in low transparency.

The opacity value of nanocomposite films could provide information on the size of the added materials dispersed within the PVA matrix during the formation of nanocomposite films. It could also be an indicator of the number of pores within the nanocomposite films [47]. If NCC had smaller particle sizes, it would fill the pores of the PVA matrix, which would induce a reduction in the light pathway, causing the nanocomposite films to have a high opacity value. 

### 3.6. Swelling Ratios of Nanocomposite Films

The swelling ratios of the nanocomposite films are shown in Figure 9. The swelling ratio decreased when NCC was added. The swelling ratio decreased from PNC0 (256%) to PNC6 (193%) (Table 3). Furthermore, the swelling ratio was controlled by the hydrophilic ability of the functional groups. The addition of NCC decreases the availability of functional groups in the matrix that can interact with water [48]. The interaction between the hydrophilic groups of NCC and PVA caused a decrease in osmotic pressure swelling, resulting in a decreased amount of water in the nanocomposite films. The increase in NCC resulted in stacking in tissue gaps and a decreased ability to absorb water [49].

The swelling ratios of the nanocomposite films decreased with increasing NCC quantities. It is clear that as NCC increases, the equilibrium solvent uptake decreases. This is due to the increased hindrance exerted by the crystals at higher loadings.

### 3.7. Analysis of Mechanical Properties

The mechanical properties were assessed using the tensile test standard, ASTM D-638 type V. Figure 10a shows the continuous loading results in nanocomposite samples during the deforming process. Figure 10b–d show the tensile strength, Young’s modulus, and elongation at the break of the nanocomposite films. 

The tensile strengths of the nanocomposite films are shown in Figure 10b. Their tensile strength increased after NCC was added to the PVA matrix. Compared to the tensile strength of PNC0 (25.3 MPa), the tensile strength of PNC2 (40.3 MPa) increased. This increase is due to the interaction of hydrogen bonds occurring between NCC and PVA, an equitable nanofiber distribution, and compatibility between the fibers and the PVA matrix [34]. The dispersion of fibers increases the number of intermolecular bonds between the nanofibers and the PVA matrix. The decreased tensile strength of nanocomposite films may occur due to the agglomeration of NCC and the uneven distribution of nanofillers in the PVA matrix [50]. The agglomerated NCC creates imperfections in the PVA matrix and causes early failure. Agglomeration decreases interfacial adhesion between NCC and the PVA matrix [51]. The decrease in tensile strength could also be due to the formation of air bubbles during the sample molding process and pore formation in nanocomposite films (blue arrow in Figure 5). 

The Young’s modulus of the nanocomposite is shown in Figure 10c. The pattern of changes in Young’s modulus, with increasing NCC content, resembles changes in tensile strength. Young’s modulus increased significantly from PNC0 (219.2 MPa) to PNC2 (581.4MPa). The increase in Young’s modulus can be attributed to the formation of the network structure generated by the interaction of the NCC and the PVA matrix via hydrogen bonding. However, in PNC3 (372.1), Young’s modulus began to decrease. The decrease in Young’s modulus can be explained by the difference in the failure strain between the NCC and PVA matrices. In other words, NCC reinforcement does not apply when the failure strain of the PVA matrix is much larger than that of the NCC.

The elongation at the break of the nanocomposite films is shown in Figure 10d. This study found that elongation at the break of nanocomposite films increased with the increasing addition of NCC in PNC2 (176.2%) and started to decrease in PNC4 (84.7%). This decrease was due to the incorporation of the NCC and PVA matrices, which may fail to form hydrogen bonds either intramolecularly or intermolecularly, thereby losing tensile strength, and hydrogen bond formation between the PVA chains is disturbed due to the presence of NCC particles [34]. Overall, it can be said that adding NCC improves the mechanical properties of the PVA matrix.

## 4. Conclusions

In this study, NCC was successfully made from tea waste, and PVA nanocomposite was added, with NCC as a filler. Observations using SEM showed that tea decomposed due to the chemical treatment to obtain cellulose. Observations using TEM showed that cellulose was broken down into NCC due to acid hydrolysis. NCC was obtained with a diameter of 6.99 ± 0.50 nm. From XRD analysis, it was found that two peak regions were amorphous and had crystalline peaks, thus indicating that NCC had semi-crystalline properties and 63.8% crystallinity. The addition of NCC affected the characteristics of the PVA nanocomposite. SEM micrographs showed the distribution of NCC on the nanocomposite films. The FTIR results showed that the peak of the O-H functional group in PVA changed after NCC was added. The Raman spectroscopy results show that the ratio of *I**_(O-H)_*/*I**_(C-H)_* is the crystallinity of the nanocomposite films after the addition of NCC, which follows the XRD results. The crystallinity increased, and subsequently, the maximum in the PNC2 composition decreased. The crystallinity affected the mechanical and thermal properties of the nanocomposite films. Mechanical properties have generally increased, as can be seen from the tensile strength and Young’s modulus after adding the NCC. The thermal properties changed after NCC was added. The melting point and glass transition temperature decreased, the degradation temperature increased, and the enthalpy also changed. The addition of NCC decreased the swelling ratio and opacity value of the nanocomposite films. Thus, tea waste can be used as a raw material in the manufacture of NCC, and as a filler in nanocomposite films.

## Figures and Tables

**Figure 1 materials-14-07154-f001:**
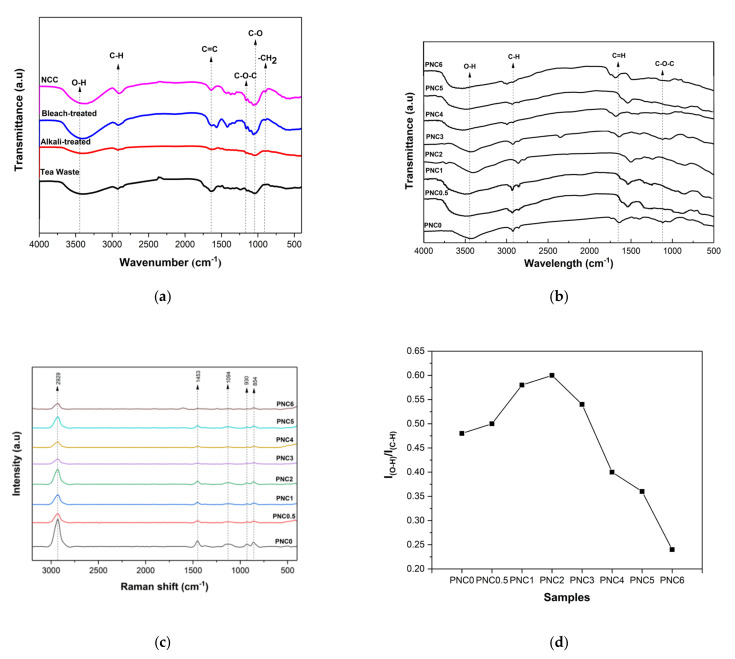
(**a**) FTIR spectra of any chemical treatment of tea waste. (**b**) FTIR spectra of PVA–NCC nanocomposite films. (**c**) Raman spectra of PVA–NCC nanocomposite films. (**d**) Intensity ratios of $I_{\left(O-H\right)}/I_{\left(C-H\right)}$ PVA–NCC nanocomposite films.

**Figure 2 materials-14-07154-f002:**
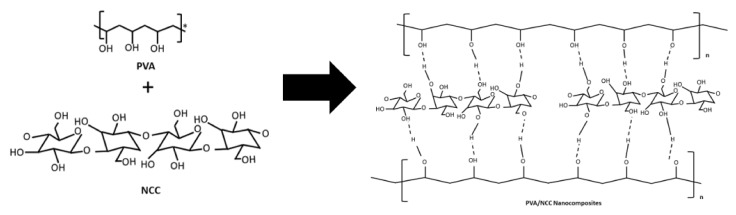
Schematic to show the intermolecular complex formed due to the hydrogen bonds between PVA and NCC.

**Figure 3 materials-14-07154-f003:**
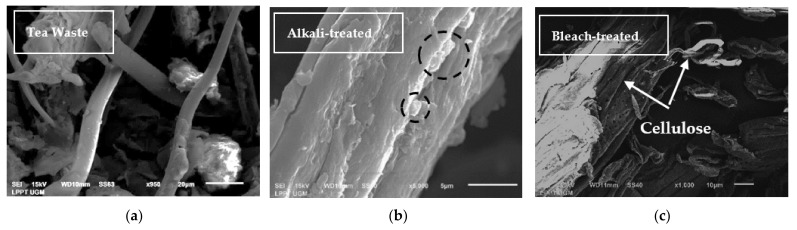
SEM micrographs of (**a**) tea waste, (**b**) alkali-treated tea waste, and (**c**) bleach-treated tea waste.

**Figure 4 materials-14-07154-f004:**
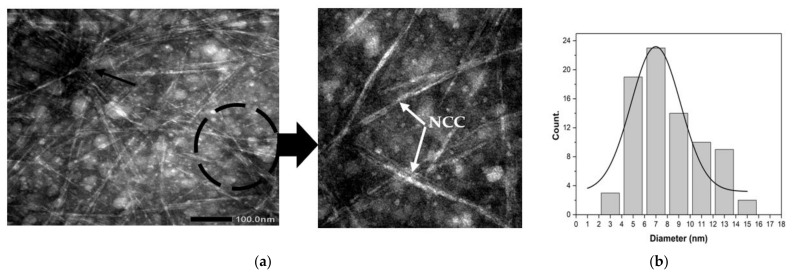
(**a**) TEM micrograph of NCC extracted from tea waste. (**b**) Diameter distribution of NCC extracted from tea waste.

**Figure 5 materials-14-07154-f005:**
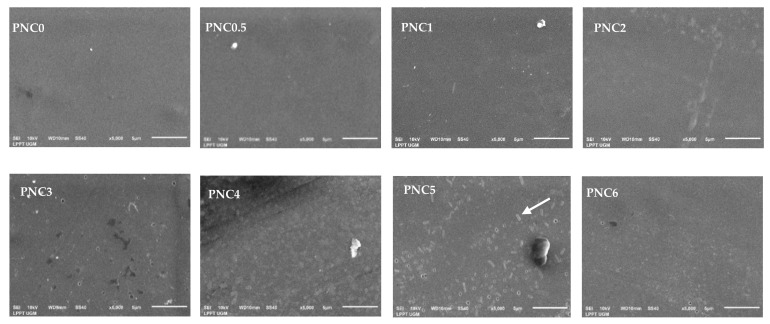
SEM micrographs of PVA–NCC nanocomposite films.

**Figure 6 materials-14-07154-f006:**
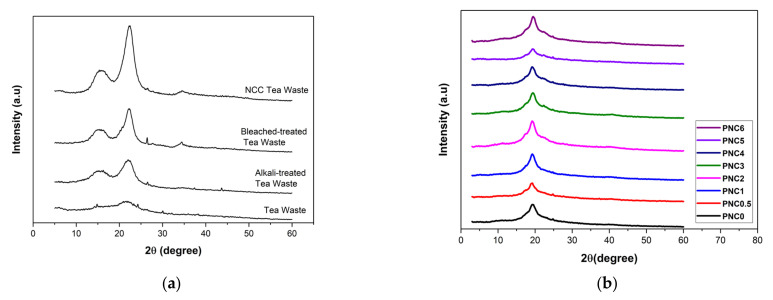
XRD diffractograms of (**a**) the chemical treatment process of tea waste and (**b**) PVA–NCC nanocomposite films.

**Figure 7 materials-14-07154-f007:**
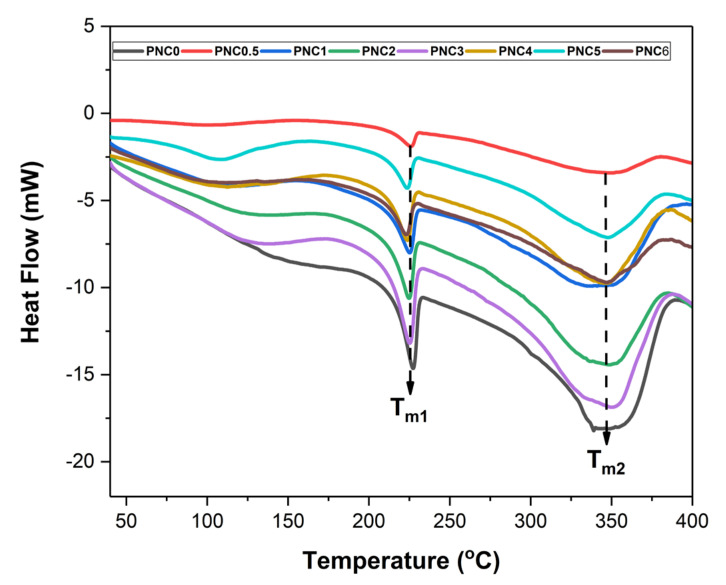
DSC curves of PVA–NCC nanocomposite films.

**Figure 8 materials-14-07154-f008:**
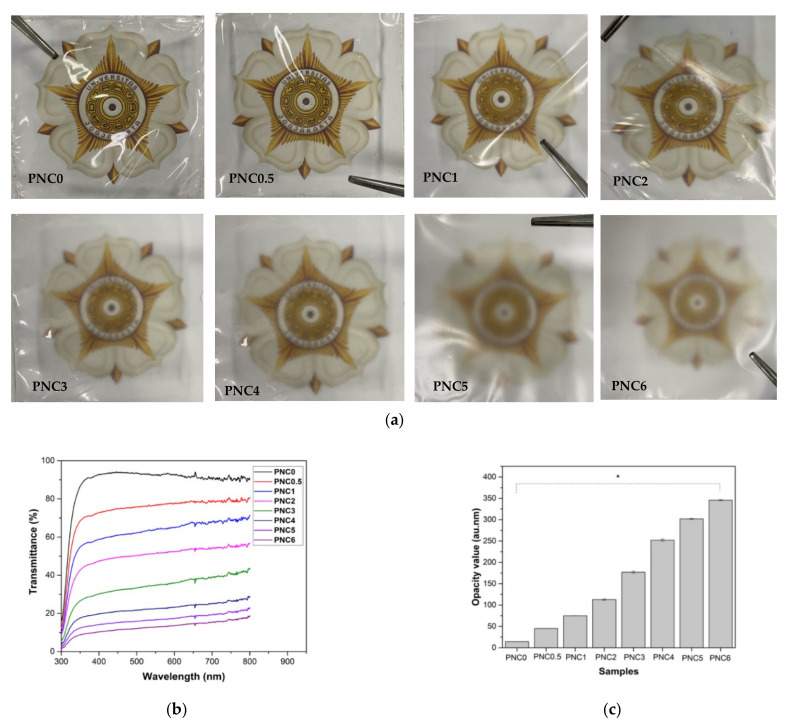
(**a**) Images of PVA–NCC nanocomposite films. (**b**) The transmittance values of PVA–NCC nanocomposite films. (**c**) The opacity values of PVA–NCC nanocomposite films (*: *p* < 0.05).

**Figure 9 materials-14-07154-f009:**
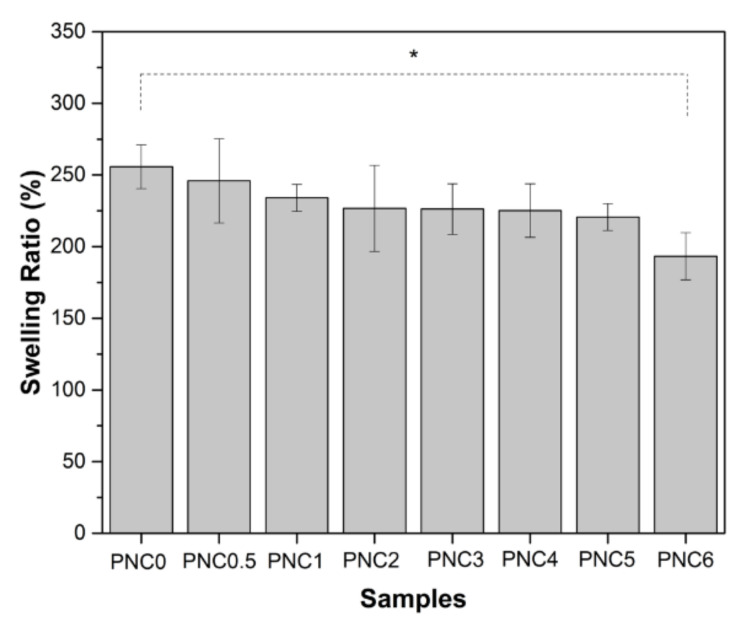
The swelling ratios of PVA–NCC nanocomposite films (*: *p* < 0.05).

**Figure 10 materials-14-07154-f010:**
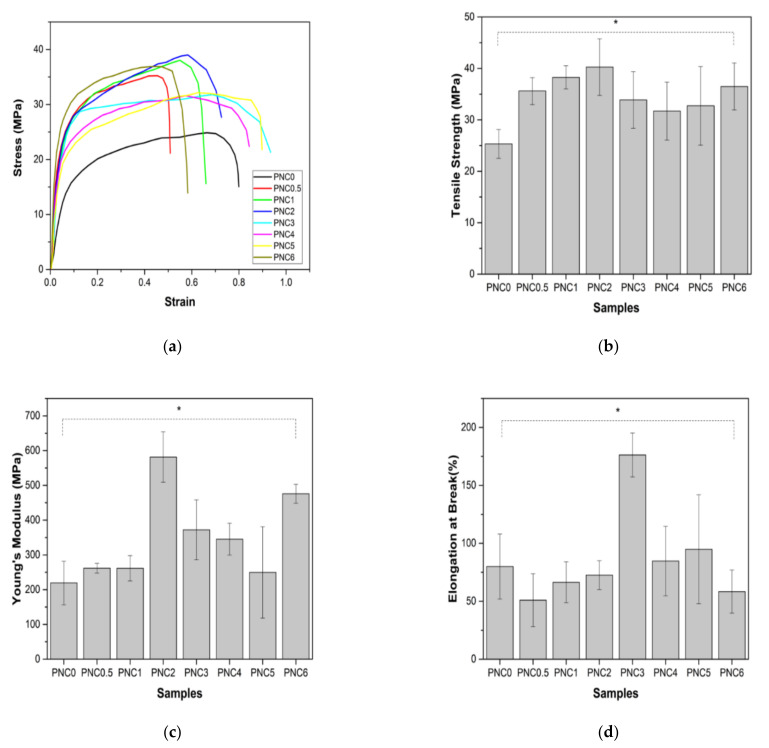
(**a**) The representative stress–strain curves of PVA–NCC nanocomposite films, (**b**) tensile strength, (**c**) Young’s modulus, and (**d**) elongation at break of PVA–NCC nanocomposite films (*: *p* < 0.05).

**Table 1 materials-14-07154-t001:** The ratios of NCC suspension and distilled water used.

Label	PNC0	PNC0.5	PNC1	PNC2	PNC3	PNC4	PNC5	PNC6
NCC suspension (mL)	0	0.5	1	2	3	4	5	6
Distilled water (mL)	10	9.5	9	8	7	6	5	4

**Table 2 materials-14-07154-t002:** The crystallinity index of tea waste at different stages of chemical treatment.

Sample	Amorphous		(002)		CI(%)
	2θ (°)	$I_{am}$	2θ (°)	$I_{002}$	
**Tea waste**	15.2	109	21.7	201	45.8
**Alkali-treated tea waste**	15.6	269	22	493	46.3
**Bleach-treated tea waste**	15.7	287	22.08	657	56.3
**NCC tea waste**	16.1	457	22.2	1263	63.8

**Table 3 materials-14-07154-t003:** The physicochemical properties of PVA–NCC nanocomposite films.

Label	CI (%)	Opacity Value (au.nm)	Swelling Ratio (%)	T_m1_ (°C)	T_m2_ (°C)	ΔH_m_ (J/g)
PNC0	29.7	14.3 ± 0.1	258.8 ± 15.2	227.3	339	52.8
PNC0.5	24.3	45.1 ± 0.1	245.9 ± 29.3	225.9	349.5	44.9
PNC1	39.5	74.9 ± 0.4	234.1 ± 9.4	225.2	338.1	73.4
PNC2	35.3	112.9 ± 1.7	226.7 ± 29.9	224.8	348.9	59.1
PNC3	34.6	177.2 ± 2.5	226.2 ± 17.7	225.3	350	58.4
PNC4	32.8	252.1 ± 2.5	225.2 ± 18.7	223.8	345.3	59.5
PNC5	29.8	301.9 ± 0.9	220.7 ± 9.4	223.7	347.9	53
PNC6	24.3	345.8 ± 1.2	193.4 ± 16.4	223.2	347.3	37

## Data Availability

Not applicable.

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
