# Peer review of "Synthesis and Physicochemical Properties of Poly(vinyl) Alcohol Nanocomposites Reinforced with Nanocrystalline Cellulose from Tea (Camellia sinensis) Waste"

_materials, 2021, doi:10.3390/ma14237154_

Round 1

Reviewer 1 Report

The authors have done a good job addressing comments from all the reviewers. Albeit that the barrier properties of the films have not been discussed. I think this manuscript is ready to publish after minor revision.

  • Please check the figure labels and make the necessary corrections (example: Figure 3b: X-axis label; Figure 7b: Y label)
  •  Please write "Perusahaan Terbatas" for PT in the manuscript.
  • It is recommended that the authors separately label the microscopic images. 

Reviewer 2 Report

In this manuscript, PVA/NCC nanocomposites were prepared by using tea waste as a kind of raw material, which was environment friendly. In my opinion, some issues should be clearer:

  • Introduction: In the first paragraph, it is presented that NCC is important. A number of methods can be employed to extract NCC from some by products. A question easily came to me that what is your contribution: new methods or new raw materials (tea waste)? This should be further discussed.
  • A number of experiments were performed. More discusses about the effects of treatments and additions of NCC on the results are needed.
  • In FTIR, the authors only correlated functional groups with the peaks. However, how the treatments affected those groups or what can the readers get from those results was not discussed.
  • Variations in the intensity ratio of I1086/I850 = I(O-H)/I(C-H) indicate the degree of crystallinity of PVA”. I did not find the 1086 and 850 peak in the manuscript. Neither did I find this equation in section 2.
  • I suggest the authors to add marks in the images in Figure 2. For example, could the cellulose, hemicellulose, and lignin be identified by SEM? It is possible to mark out the NCCs by circles in Figure 2(b)?
  • There was also a peak in 100-150°C in Figure 4. What was that?

Author Response

Point 1 : Introduction: In the first paragraph, it is presented that NCC is important. A number of methods can be employed to extract NCC from some by products. A question easily came to me that what is your contribution: new methods or new raw materials (tea waste)? This should be further discussed.

Response 1 : The author has corrected the introduction in the manuscript.

Point 2 : A number of experiments were performed. More discusses about the effects of treatments and additions of NCC on the results are needed.

Response 2 : The author has added a discussion about adding NCC to each experiment.

Point 3 :In FTIR, the authors only correlated functional groups with the peaks. However, how the treatments affected those groups or what can the readers get from those results was not discussed.

Response 3 : The author has added how the treatment affected those groups.

Point 4 :“Variations in the intensity ratio of I1086/I850 = I(O-H)/I(C-H) indicate the degree of crystallinity of PVA”. I did not find the 1086 and 850 peak in the manuscript. Neither did I find this equation in section 2.

Response 4 : The equation in Section 2 has been added. To determine the intensity ratio shows the crystallinity of PVA the author has replaced that part because, in this study, the peak intensity of OH appeared at 1094 and the peak intensity of CH at 854, so the authors corrected statement in the results Section 3.1 and Figure 1d.

Point 5 : I suggest the authors to add marks in the images in Figure 2. For example, could the cellulose, hemicellulose, and lignin be identified by SEM? It is possible to mark out the NCCs by circles in Figure 2(b)?

Response 5 : The author has added marks in the image in Fig. 2 and marked out the NCCs by circles in Fig. 2b.

Point 6 :There was also a peak in 100-150°C in Figure 4. What was that?

Response 6 : The peak at 100-150°C shows an endothermic peak, which indicates that the heat energy was used to evaporate the water contained in the nanocomposites.

Reviewer 3 Report

The work entitled "Synthesis and physicochemical properties of poly(vinyl) alco hol (PVA) nanocomposites reinforced nanocrystalline cellulose (NCC) from tea (Camellia sinensis) waste" needs a major revision.

Comments:

  1. The title should be modified. It should be "Synthesis and physicochemical properties of poly(vinyl) alcohol (PVA) nanocomposites reinforced with nanocrystalline cellulose from tea (Camellia sinensis) waste".
  2. Introduction is poor. The authors have to have a good literature survey and enhance the quality of this section.
  3. Please add a schematic to show the intermolecular complex formed due to the hydrogen bonds 4. between PVA and NCC.
  4. Figure 1.d, there is an increase and then a decrease in the intensity ratio. Why? you need to discuss this in the manuscript.
  5. Tg: The authors claim that "In PNC0, the Tg was 336 76.54°C, and it decreased after the addition of NCC. This is because the mobility of the 337 PVA chain is limited by the absorption style of the NCC." This is incorrect. If a nanofiller hinders movement of chains, tg would increase.
  6. "The decreased tensile strength of nanocomposites may occur due to the clumping (agglomeration) of NCC and the uneven". You should discuss more and cite relevant references from literature for this phenomenon. You can use the following refs: a) Improving the thermal and mechanical properties of poly(vinyl butyral) through the incorporation of acid-treated single-walled carbon nanotubes b) Poly(vinyl chloride) PVC / single wall carbon nanotubes (SWCNT) composites: Investigation of mechanical and thermal characteristics.
  7. Table 4 and fig 6b-d report the same data. Please remove either the table or the figure 6b-d.
  8. There is an increase in modulus and strength of PNC6 and elongation at break of PNC3. How do you justify these observations?
  9. English is not acceptable and there are lots of errors. The manuscript needs to be re-writhen by a native speaker. Some cases are listed below:

-were 154 shown in Fig

-sspectra

-any treatment chemical

-Fig. 2.b show

-Crystallinity nanocomposites

Round 2

Reviewer 2 Report

Most of my concerns solved by the authors. However, the points 5 and 6 need further revision.

For point 5, circles are not found neither in Figure 2b nor Figure 3a. In addition, I am not sure how to clearly distinguish NCCs from Figure 4(a), if NCCs were marked by red circles in Figure 4(a). The NCCs seem to be not very clear in the figure.

For point 6, in addition to the reply to me, discussions should also be added in the manuscript to avoid any potential confusions from readers.

Reviewer 3 Report

Dear editor,

The authors of the manuscript entitled "Synthesis and physicochemical properties of poly(vinyl) alcohol nanocomposites reinforced with nanocrystalline cellulose from tea (Camellia sinensis) waste" have addressed the comments. So the work should be accepted for publication in Materials.

Author Response

This manuscript is a resubmission of an earlier submission. The following is a list of the peer review reports and author responses from that submission.

Round 1

Reviewer 1 Report

In this work, the authors isolated nanocellulose through different chemical treatments, in order to use them in the preparation of poly (vinyl alcohol) / nanocellulose nanocomposites. The authors seem to have worked a lot; however, it is difficult to evaluate the work due to the high number of errors and also to the lack of discussion.

For example:

In abstract, introduction, materials, extraction of NCC and preparation of PVA/NCC nanocomposites section

Line 12. What type of hydrolysis?

Line 16. "...characterization..."? Maybe authors are referring to "properties"

Line 18: “decrease transparency” has the same meaning of “increase in opacity value”. Both are redundant in the same sentence

Line 21. It is somewhat daring to propose composites for tissue engineering without additional characterizations, such as in vitro biocompatibility

Line 42. What want authors to mean with "versatility"?

Line 47. Very high quantity for a nanoentity. Nanocomposites usually contains less than 15 wt%

Line 54. What wants to mean with "group function"?

Line 59. Define the significance of PT

Line 63. Add the country

Line 66. Add the unity

Line 66. In the next sentence "...then initialized using a solution of 6% (b/v) NaOH...", what is the meaning of initialized? Please correct (b/v)

Line 68. "...the solution was neutralized using distilled water...". "neutralized"?

Maybe, "whased with distilled water until neutral pH was reach"?

Line 71. "compared"???

Line 74. What is the meaning of the reference?

Line 75. In which ratio?

Line 78. Please indicate the solvent

Line 79. "...the mixture was formed and dried...", "mixture"? Maybe authors are referring to "nanocomposite film"?

Line 80. I don't understand in the sentence the meaning of "comparison variation"

Table 1. What is the NCC concentration in the suspension?

In results section

In the discussion of FTIR results

The authors attempt to assign the observed peaks to the different functional groups. Some of the assignments are questionable. But, the most questionable thing about this work is that there is no discussion of the results obtained. For example, Figure 1a shows the spectra of (a) tea waste, (b) alkali treated tea waste, (c) bleached tea waste, and (d) NCC, where variations in bands can be observe. However, the manuscript does not discuss the effects of the chemical treatments on the elimination of hemicellulose or lignin, nor the relationship between the removed components and the appearance or disappearance, or displacement of the characteristic bands. I consider that the results are not discussed accordingly in the manuscript. To better document these effects, I suggest to read the manuscript http://dx.doi.org/10.1016/j.indcrop.2014.02.014

Therefore, I suggest that the manuscript be reviewed and corrected according to (i) the writing, so that the sentences and paragraphs contain the information that the authors want to convey, and (ii) the discussion of the results, that the authors justify the treatments and procedures carried out with the materials obtained and that are supported by the results obtained from the characterization. This information must reach to the readers and scientific community clearly.

Reviewer 2 Report

The submitted manuscript deals with the isolation of nanocrystalline cellulose (NCC) from tea waste and its use as filler in PVA nanocomposites. According to the Introduction, NCC are added to the PVA matrix to reduce its swelling in wet environments which results in mechanical property and oxygen barrier deterioration.

The use of NCC as filler/reinforcement of PVA has been extensively studied. In this context, the novelty of this article is relatively low since it could only rely in the NCC source used (tea waste). Characterization is also the usual one. Moreover, and despite it was mentioned in the introduction section as one of the matrix disadvantages to be improved,  the effect of NCC addition on the composite oxygen barrier properties was not measured.

Results analysis is very confusing since language translation seems to have resulted in very uncommon English use, and a lot of non-proper terms and sentences which are not those conventionally used in this type of articles are found. Besides, several mistakes were identified (see details below) and discussion is often limited to a collection of comments from the literature which do not always apply.

Overall, I consider this article not suitable for publication in Polymers.

Some details to be considered in a later submission:

  • Introduction: no state of the arte on PVA-NCC composites is given.
  • Line 26: please revise. This line is confusing since in the preparation of NCC most amorphous regions of cellulose fibrils are expected to be selectively removed.
  • Line 42: please revise this claim on PVA’s nanostructure.
  • Line 47: please complete. Much lower NCC concentrations have proved to have the same effect.
  • English needs thorough revision by a native speaker (just some examples: lines 53-54, 62, 80-81, 83, 86, ). Verb tenses need to be standardized (e.g. last paragraph of the introduction section).
  • Why did authors choose a 100% hydrolyzed PVA?
  • It is quite uncommon that Methods start with “The…”.
  • Line 66: what do authors mean with “initialized”?
  • Line 71: what do authors mean with “compared”?
  • Section 2.2 needs to be reorganized and the title should consider that not only NCC extraction method is reported (actually only described in lines 75-76) but also cellulose isolation from tea waste is described in detail.
  • Section 2.3. The use of “solvent” for the aqueous suspension containing the NCC is quite uncommon.
  • Line 78: please verify the concentration of the solution with the mass and volume values reported.
  • Line 79: what do authors mean with “formed”?
  • Line 81: what do authors mean with “comparison variation”?
  • Line 86: why should NCC composition change upon addition into the PVA films?
  • Section 2.4.1: Why did authors prepare KBr-tablets and not directly assay the PVA nanocomposite films?
  • Section 2.4.3. Authors should specify which “influence” they refer to.
  • In general the style of the Methods subsections should be standardized (verbal tenses, description of samples assayed (e.g. “The influence of each NCC manufacturing process and the effect of NCC addition to nanocomposites were analyzed using a…” in the SEM description; “The crystallinity of the sample was measured…” for XRD subsection, etc), subsection titles (analytical technique assayed vs. properties measured, i.e. “Fourier transform infrared spectroscopy (FTIR)” vs “Analysis of thermal properties”, etc), use of abbreviations in titles subsections, use of dots at the end of titles, and son on)
  • Line 79: what do authors mean with “the transmittance value was measured below the transmittance spectrum region”?
  • Section 2.4.6: Why would have authors had to “mold” the samples for UV-Vis analysis if they already had films?
  • Line 140. ”FTIR was used to assess the cellulose and to observe the effect of adding NCC to the bonds in PVA.” Please thoroughly revise the article and rephrase sentences using specific language.  
  • Lines 141-142: “In Figure 1.a, we can see the effect of chemical processes in the manufacture of NCC in the spectra produced by FTIR. At a peak of about 3650 cm-1–3200 cm-1, there is an O-H stretching produced by the –OH group in cellulose molecules”. Re-phrasing of sentences of this type is essential.
  • Line 143-144: “At 2922 cm-1, 143 there is an attribute of the C-H stretching …”. The same as before.
  • Line 146-147: “The C-O stretching vibra-146 tion at 1300 cm-1–1000 cm-1 is produced by a very strong group of alcohols”. The same as before.
  • Line 140 and others: please avoid the use of “we”, “our”, “us”, etc.
  • Lines 150-152: How do authors identify amorphous and crystalline regions in the FTIR spectra?
  • Line 159: “At a peak of 1737 cm-1, there are the C=O and C-O stretching of acetate group residues in the PVA matrix”. How is this possible if the PVA used is 100% hydrolyzed?
  • FTIR analysis is very poor since only description of bands in uncommon language is given and no reference to which sample is being described is given. Nor systematic comparison among the samples with different chemical treatments is given.
  • Figure 1. Samples labeling should be explained.
  • Line 178: What is the explanation for this?
  • SEM analysis is repetitive and includes conclusions that cannot drawn from the images included. Unconventional language is also used.
  • Lines 202-204 contradict line 26.
  • Lines 207-209. This is not evidenced in Figure 2b.
  • Figure 2: Is this NCC? Which length are they? The Figure should show it.
  • Lines 221-222: this is not supported by the data of this section.
  • Table 2 (XRD). Segal equation is not valid for uncompletely delignified samples.
  • Description of XRD nanocomposite data is all mixed up and contradictory conclusions are driven in different sentences.
  • Conclusions: ideas are repeated many times and some of them are not supported by data.

Reviewer 3 Report

This work presents "Synthesis and Physicochemical Properties of Poly(vinyl) alcohol (PVA) Nanocomposites Reinforced Nanocrystalline Cellulose (NCC) from Tea (Camellia sinensis) Waste". Application of cellulose nanomaterials for improving mechanical and barrier properties of nanocomposites for packaging or tissue engineering applications is an interesting topic for both academic research areas and industries. Preparation of CNC from tea waste is important and shows the potential application of agricultural residues for production of more green composites and products. This manuscript is recommended to be published after including and addressing the below listed comments with major corrections.

- The authors should eliminate the current grammatical and punctuation mark errors and also confirm the correct scientific English.

- Since the authors suggested that the fabricated nanocomposite can be applied for packaging, it would be very interesting if the authors check the water contact angle, water vapor permeation, and oxygen permeation.

- The authors should write the complete terms of all abbreviations (including the instruments) before the first use in the abstract and main manuscript. For example sodium hypochlorite in line 62. What does PT in line 59 or PNC stand for?

- The authors should clearly explain the innovation and importance of their work on the introduction of the manuscript. They should justify the value of the work and compare their work with previously similar published papers which applied cellulose nanomaterials from agricultural wastes to improve the PVA-based composites. For example: "Pereira et al., Carbohydrate Polymers, 112 (2014), pp.165-172", or "Asad et al., Carbohydrate Polymers, 191 (2018), pp.103-111". The introduction section needs to be elaborated.

- The tea waste does not contain 87.9% cellulose. The cellulose content of tea wastes was previously reported to be around 16.2% (Rahman et al., Polymers 2017, 9, 588) or 29.42% (Tutus et al., (2015). “Evaluation of tea wastes” BioResources 10(3), 5407-5416) depending on the source and extraction conditions. Based on the results reported by Rahman et al., Polymers 2017, 9, 588, The value 87.9% is related to cellulose content of tea wastes after bleaching. Please correct this value in the revised manuscript.

- Please check the reference 4 in line 34 (Jia, Y.; Hu, C.; Shi, P.; Xu, Q.; Zhu, W.; Liu, R. Effects of Cellulose Nanofibrils/Graphene Oxide Hybrid Nanofiller in PVA 370 Nanocomposites. Int. J. Biol. Macromol. 2020, 161, 223–230. https://doi.org/10.1016/j.ijbiomac.2020.06.013) which seems to be irrelevant.

- The authors should check all the references as Ref.4 seems to be irrelevant.

- Please briefly explain the acid hydrolysis process in line 75.

- Instead of reporting the ratio of CNC to PVA in v/v, it is recommended the authors report the addition amount of NCC in the nanocomposite based on the wt. % relative to PVA or parts per hundred of dried PVA (pph).

- Please provide more information for microscopy process like concentration of the CNC or substrate.

- Line 170: "The peaks that appear at 2929 cm-1, 1453 cm-1, 930 cm-1, and 854 cm-1 are characteristic of PVA". Line 175-177: "The peak at 1094 cm-1 is associated with the PVA amorphous regions, and this peak is influenced by the addition of NCC". Please provide relevant reference.

- The authors should label each image or graph separately.

- What do the authors mean by more clean fibers after alkaline treatment or bleaching process?   SEM images of original “tea waste” fibers and the “alkali-treated” fibers are not comparable and it’s not possible to observe more “clean” fibers.

- The scale of the SEM images are not clear, please provide a self-made scale values.

- In figure 2, please change the “bleached-treated” with “bleached fibers” or something else.

- SEM images of “Bleached fibers“ does not show fibrous materials clearly. Please provide a better image.

- Line 11 in Abstract “To obtain NCC, a chemical process is carried out in the form of hydrolysis with alkaline treatment, hydrolysis with bleaching and hydrolysis”, do the author mean: ”a chemical process is carried out in the form of alkaline treatment, followed by a bleaching and acid hydrolysis”? Please check the sentence and make the proper changes.

- Please check line 206: “The shape of the NCC particles is thought to be that is thought to be due to an imperfect cooling process (quenching) at the time of cessation of the acid hydrolysis process”